# Controlling the selectivity of the hydrogenolysis of polyamides catalysed by ceria-supported metal nanoparticles

XinBang Wu [1], Wei-Tse Lee[1], Roland C. Turnell-Ritson [1], Pauline C. L. Delannoi[1], Kun-Han Lin [2] ✉ & Paul J. Dyson [1] ✉

Catalytic hydrogenolysis is a promising approach to transform waste plastic into valuable chemicals. However, the transformation of N-containing polymers, such as polyamides (i.e. nylon), remains under-investigated, particularly by heterogeneous catalysis. Here, we demonstrate the hydrogenolysis of various polyamides catalysed by platinum-group metal nanoparticles supported on $CeO_2$. $Ru/CeO_2$ and $Pt/CeO_2$ are both highly active but display different selectivity; $Ru/CeO_2$ is selective for the conversion of all polyamides into water, ammonia, and methane, whereas $Pt/CeO_2$ yields hydrocarbons retaining the carbon backbone of the parent polyamide. Density functional theory computations illustrate that Pt nanoparticles require higher activation energy for carbon−carbon bond cleavage than Ru nanoparticles, rationalising the observed selectivity. The high activity and product selectivity of both catalysts was maintained when converting real-world polyamide products, such as fishing net. This study provides a mechanistic basis for heterogeneously catalysed polyamide hydrogenolysis, and a new approach to the valorisation of polyamide containing waste.

Polyamides (PAs) are a class of plastic with myriad uses in the automotive, building and construction, electronics, textiles and fishing industries[1]. Over 8 million tonnes of PAs are produced each year, with this number set to increase to over 10 million by 2027[2,3]. The range of applications is a result of the impressive and tuneable properties of PA-based materials, which have their origins in the chemistry of the amide bond. The amide linkage is stronger than its oxygen-only counterpart, the ester[4], but an amide can act as both hydrogen bond donor and acceptor, creating crosslinked regions that impart immense strength to the material – enough to stop a bullet, in the case of Kevlar[5,6]. However, it is this very durability that is contributing to an environmental crisis on a global scale[7]. Improper disposal of PAs and other plastics leads to their accumulation in ecosystems, where toxic additives and break-down products leach into the environment over many years[8,9]. One of the most widely publicised reservoirs of plastic waste, the 'Great Pacific Garbage Patch', is primarily composed of fishing industry waste, of which ropes and netting made from PAs constitute a sizeable fraction[10,11].

Research into end-of-life processing of PA waste is currently limited (Supplementary Table 1). Techniques explored include pyrolysis[12,13], hydrolysis[14,15], acid- or base-catalysed depolymerisation[16,17], ionic-liquid catalysed solvolysis[18], metal-complex catalysed hydrogenation[19,20] and hydrogenative ammonolysis[21,22,23]. Two of these studies investigated the conversion of PA-blends (containing a mixture of different PAs) or PA composites[13,21], which contain PA blended with other materials, such as carbon fibre or glass. Real-world PA waste typically contains not only pure polymers, but also other materials (plasticisers, reinforcers, flame retardants, etc.)[24,25] making processing methods that can handle waste streams of varying composition a pressing concern[26,27]. Common PA composites reinforced by glass- or carbon-fibres are mainly used as gears for machinery or automotive parts[28–30].

[1]Institute of Chemical Sciences and Engineering, Swiss Federal Institute of Technology Lausanne (EPFL), Lausanne, Switzerland. [2]Department of Chemical Engineering, National Tsing Hua University (NTHU), Hsinchu, Taiwan. ✉e-mail: kunhan.lin@mx.nthu.edu.tw; paul.dyson@epfl.ch

Depolymerisation of plastic waste via hydrogenolysis has received significant interest, with numerous recent studies demonstrating the successful conversion of polyolefins into short-chain hydrocarbons[31–40]. At reaction temperatures between 200 and 250 °C, the conversion of polyolefins into liquid products can be achieved[35–40]. Polyolefin hydrogenolysis is challenging due to the strength of the carbon–carbon σ bond (C−C), but these systems are simplified as only the cleavage of a single type of bond needs to be considered. The reaction rate and product selectivity may be drastically different for polymers that contain both C−C and carbon-heteroatom (specifically C−O and C−N) bonds. N-atoms in particular are problematic, as they can poison commonly used catalysts and decrease the rate of hydrogenolysis[41]. Furthermore, a broad range of functionalised compounds might be produced from the incomplete cleavage of carbon−heteroatom bonds, necessitating undesirable purification steps. To improve the product selectivity when transforming heteroatom-containing polymers, selective bond cleavage can be considered. For instance, selective alkoxy C−O bond hydrogenolysis of polyesters can produce high yields of the valuable dicarboxylic acid monomer, terephthalic acid[42].

Despite the various studies that focus on the design of heterogeneous catalysts for selective C−O bond hydrogenolysis[42–44], there have been limited studies exploring the cleavage of carbon−heteroatom bonds in PAs. However, a supported Ru catalyst was used to confirm the homogeneity of a Ru-complex used to catalyse PA hydrogenation[20]. Herein, we report the study of the catalytic hydrogenolysis of N-hexylhexamide (which serves as a model substrate for PA-6, also known as nylon-6, one of the most produced PAs[2]) using platinum-group metal (Ru, Rh, Pd, Ir, and Pt) nanoparticles (NPs) supported on ceria (CeO$_2$). Following identification of the most active and selective catalysts, the catalysts were evaluated in the depolymerisation pure PAs, PA blends, PA composites and real-world sources of waste PA. Insights into the mechanisms underpinning the hydrogenolysis reactions explain the differences in product selectivity of the catalysts.

## Results

### Catalyst synthesis and characterisation

M/CeO$_2$ (M = Ru, Rh, Pd, Ir or Pt) catalysts were prepared via incipient wetness impregnation using metal chloride precursors loaded onto CeO$_2$ nanopowder (particle size <25 nm)[45]. This step was followed by low-temperature annealing, excess metal precursor removal and high-temperature calcination (see Methods for full details). The removal of excess metal precursor after the low-temperature annealing step prevents metal aggregation into larger NPs, facilitating good dispersion[46].

Scanning transmission electron microscopy (STEM) of the M/CeO$_2$ catalysts reveals that the metal-oxide NPs generally comprise of particles <10 nm in diameter (Fig. 1). The uniform morphology of the CeO$_2$ support may be observed for all the M/CeO$_2$ catalysts, exhibiting smooth surfaces with well-defined particle boundaries. Ru and Pd NPs show a higher tendency to agglomerate than the Rh, Ir and Pt NPs, resulting in larger average NP diameters (Ru = 8.9 ± 2.8 nm, Pd = 7.7 ± 2.4 nm, vs. Rh = 1.9 ± 0.3 nm, Ir = 1.1 ± 0.2 nm, Pt = 0.9 ± 0.2 nm; Fig. 1f). However, results from the catalyst screening showed that the slight difference in the size of the NPs did not significantly influence relative catalytic activity between each noble metal species (see below).

Powder X-ray diffraction (PXRD) was performed on the M/CeO$_2$ catalysts, and the cubic CeO$_2$ diffraction pattern was observed in every case. Additionally, weak intensity peaks corresponding to the noble metal oxides were observed for Ru/CeO$_2$, Pd/CeO$_2$, Ir/CeO$_2$ and Pt/CeO$_2$ (Supplementary Fig. 1). Additionally, X-ray photoemission spectroscopy (XPS) was employed to analyse the surface composition of the M/CeO$_2$ catalysts (Supplementary Fig. 2), indicating that the noble metal species were mostly metal oxides, with some metal chlorides species present in Ru/CeO$_2$, Pd/CeO$_2$ and Ir/CeO$_2$ (see detailed assignments in Supplementary Table 2) The metal-oxide NPs are likely anchored through surface oxides, i.e. M-O-Ce at the interface[47–49]. The actual metal loading in M/CeO$_2$ was determined by inductively coupled plasma mass spectrometry (ICP-MS), and ranges from 3.7 wt.% to 5.2 wt.% (Supplementary Table 3).

### Catalyst screening

N-hexylhexamide was used as a model for the polyamide PA-6, as both contain one amide linkage between two hexyl-chains. The activities of the M/CeO$_2$ catalysts were screened for N-hexylhexanamide conversion and product distribution at a reaction temperature of 325 °C and (initial) hydrogen pressure of 50 bar. The products were analysed using gas chromatography (GC) and are shown in Fig. 2a. Control reactions were performed to determine the thermal stability of N-hexylhexanamide under the reaction conditions, which undergoes low levels (6%) of thermal decomposition in the absence of catalyst. In the presence of CeO$_2$, 24% conversion to non-alkane products is observed (Supplementary Table 4). These observations confirm that hydrogenolysis of PAs to alkanes with the M/CeO$_2$ catalysts is primarily due to the activity of the supported metal NPs[50]. The catalyst screening studies reveal that the predominant products are linear alkanes. In addition, the hydrogenolysis of C−O and C−N bonds leads to the formation of water and ammonia, both observed by NMR. To quantify the amount of ammonia produced, a colorimetric analysis based on

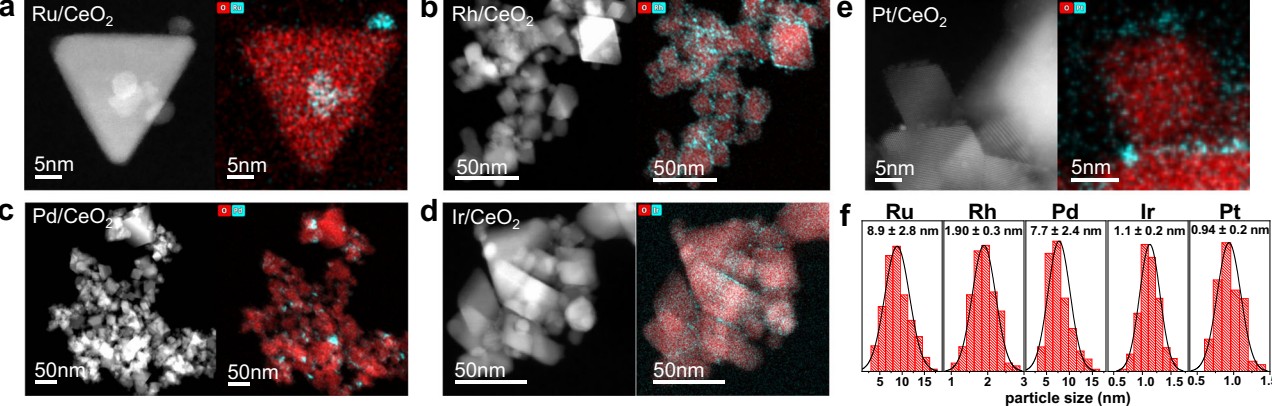

**Fig. 1 | STEM-HAADF and EDX analysis of M/CeO$_2$ (M = Ru, Rh, Pd, Ir or Pt) catalysts.** STEM images of (**a**) Ru/CeO$_2$, (**b**) Rh/CeO$_2$, (**c**) Pd/CeO$_2$, (**d**) Ir/CeO$_2$ and (**e**) Pt/CeO$_2$ (left-hand side: STEM-HAADF; right-hand side: STEM-EDX). **f** Particle size distribution of metal NPs of M/CeO$_2$ catalysts, the means ± standard deviations (particle counts ≥200) are shown.

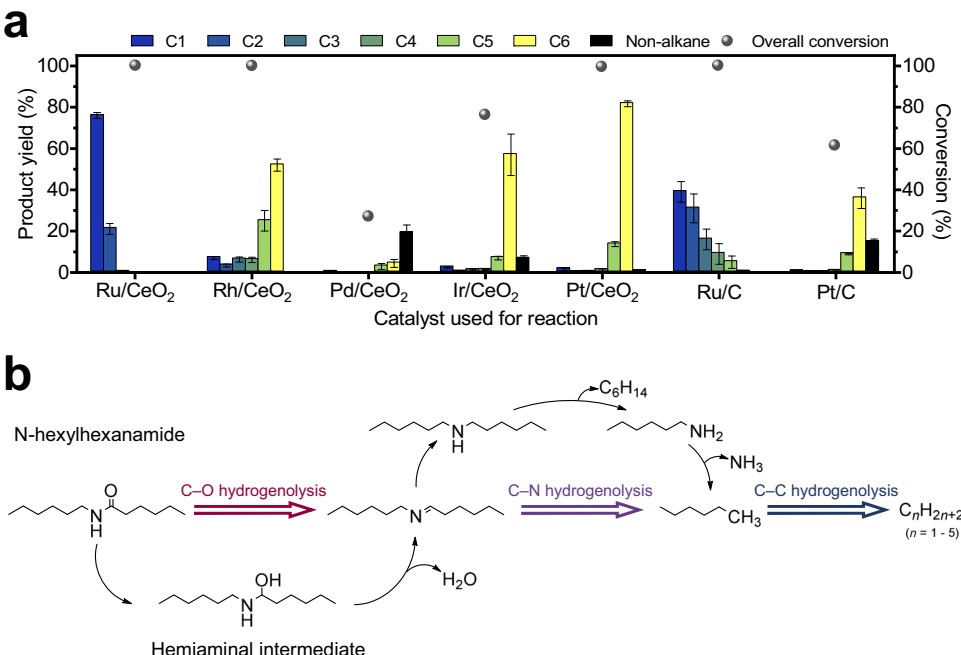

**Fig. 2 | Conversion of *N*-hexylhexanamide with the M/CeO$_2$ (M = Ru, Rh, Pd, Ir or Pt) catalysts.** Reaction conditions: *N*-hexylhexanamide (250 mg), catalyst (25 mg, 0.5 wt% of metal) and H$_2$ (50 bar) at 325 °C for 5 h. All reactions were repeated three times and the reported values represent the mean percentage yield with the corresponding standard error ($n \geq 3$). **a** Product distribution from the hydrogenolysis of *N*-hexylhexanamide with M/CeO$_2$ and commercial carbon-based catalysts. Non-alkane products are given in Supplementary Table 4. **b** Reaction pathway of alkane formation from the consecutive C–O, C–N and C–C hydrogenolysis of *N*-hexylhexanamide. Initial C–O cleavage via a hemiaminal intermediate is considered to be the predominant reaction pathway for most heterogeneous amide hydrogenation catalysts, with the absence of O-containing products in the product distribution supporting this mechanism[76].

an indophenol assay was conducted. The results demonstrate that a significant portion of the nitrogen content in the amide was converted into ammonia, with the Ru/CeO$_2$ and Pt/CeO$_2$ catalysed reactions showing the highest (near-quantitative) conversions (Supplementary Fig. 9).

The Ru/CeO$_2$ catalyst was found to be the most active towards hydrogenolysis, displaying complete conversion of *N*-hexylhexanamide with a 76% yield of methane. This result aligns with prior studies on polyolefin conversion, where Ru is consistently recognised as the most active of the platinum group metals for C–C hydrogenolysis[32,37,40]. Furthermore, the high selectivity towards methane observed in this reaction is consistent with previous findings on the hydrogenolysis of polyolefins at temperatures above 300 °C[31]. Additionally, the Rh/CeO$_2$, Ir/CeO$_2$ and Pt/CeO$_2$ catalysts give conversions of *N*-hexylhexanamide of >99, 99 and 98%, respectively. The selectivity towards alkane products varies between these three catalysts, highlighting differences in their hydrogenolysis activity. Despite *n*-hexane being the predominant hydrocarbon product for all three catalysts, the varying activity among Rh, Ir and Pt in activating C–C bond cleavage results in different yields of *n*-hexane, i.e. 52, 57 and 82%, respectively. It has been proposed previously that C–C bond hydrogenolysis occurs only at the chain end[31], and this hypothesis is supported by the observation of methane and *n*-pentane as the main components of the non-*n*-hexane products. The Pt/CeO$_2$ catalyst was observed to afford the highest yield of *n*-hexane, suggesting that it is the least able to cleave C-C bonds. Nevertheless, it effectively cleaves the C–O and C–N bonds, which results in the hydrogenolysis of the amide linkage.

The Pd/CeO$_2$ catalyst was the least active of the M/CeO$_2$ hydrogenolysis catalysts, resulting in a low overall conversion of *N*-hexylhexanamide of 27%. The product distribution consisted of alkanes (8%) and non-alkanes (19%), the latter comprising hexylamine (1%), dihexylamine (9%), trihexylamine (3%) and *N,N*-dihexylhexanamide (6%)

(Supplementary Fig. 8). The lower activity of Pd/CeO$_2$ may be attributed to the higher proportion of PdCl$_2$ species, as revealed by XPS analysis (Supplementary Fig. 2c). Chloride is known to strongly bind to the active catalytic sites, which inhibits hydrogenolysis and causes deactivation of the catalyst[32]. However, the examination of non-alkane products holds significance for understanding the mechanism of the hydrogenolysis reaction. Besides the formation of amines, the absence of any alcohols or alcohol derivatives, indicates the absence of alcohol formation through direct amide C–N bond cleavage. This suggests that the C–O bond cleavage occurs first, and is followed by dehydration, producing water and dihexylamine as the major hydrocarbon product (Fig. 2b), which have been already proposed as a possible mechanism for the hydrogenation of secondary amides[51]. Subsequently, dihexylamine can undergo C–N bond cleavage on one or both sides of the molecule, yielding *n*-hexane and hexylamine or *n*-hexane and ammonia as products, respectively. The experimental results across the full suite of M/CeO$_2$ catalysts suggest that the intrinsic activity of the metal NPs in the catalyst does not impact the order of C–O and C–N bond hydrogenolysis relative to C–C bond hydrogenolysis. This is largely due to the difference in bond strengths (i.e. C–C > C–N > C–O σ > C–O π), as weaker bonds are more susceptible to hydrogenolysis[52]. The formation of tertiary amines and amides, such as trihexylamine and *N,N*-dihexylhexanamide, presumably originate from the processes of disproportionation and transalkylation of the starting amide or intermediate amine, which compete with hydrogenolysis[53]. The catalyst screening results provide a ranking of the amide hydrogenolysis activity of the metals, with Ru/CeO$_2$ showing the highest activity, followed by Rh/CeO$_2$, Pt/CeO$_2$ and Ir/CeO$_2$, and with Pd/CeO$_2$ being the least active, which is consistent with a previous study[54]. However, given the lower propensity for Pt to cleave C–C bonds, it has a greater selectivity towards hexane. Based on these results, Ru/CeO$_2$ and Pt/CeO$_2$ were chosen for further investigation.

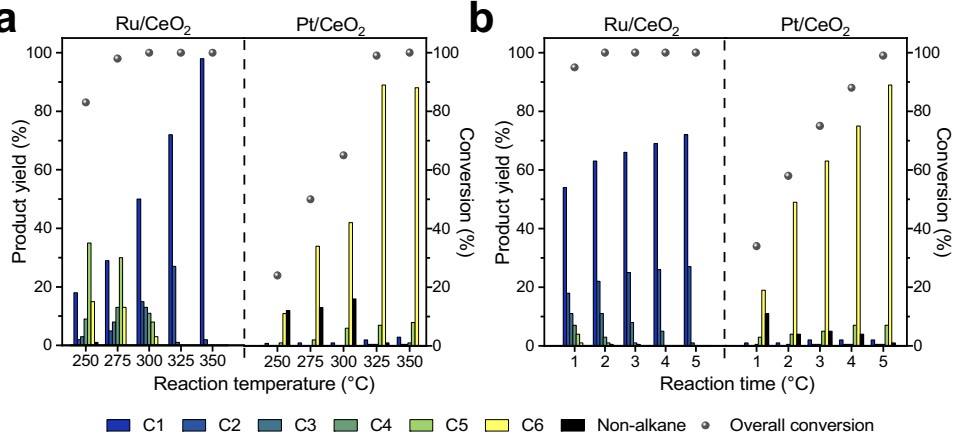

**Fig. 3 | Temperature and time dependence of *N*-hexylhexanamide hydrogenolysis using the Ru/CeO₂ and Pt/CeO₂ catalysts. a** The effect of reaction temperature on the conversion after 5 h. **b** The effect of reaction time on the conversion of *N*-hexylhexanamide at 325 °C. Reaction conditions: *N*-hexylhexanamide (250 mg), catalyst (25 mg, 0.5 wt% of metal) and H₂ (50 bar). Supplementary Tables 5–7 provide detailed product yields.

## Effect of the support

A comparison of M/CeO₂ (M = Ru and Pt) and commercially available M/C catalysts (M = Ru and Pt) was conducted to evaluate the influence of the support material. Under the same conditions (*N*-hexylhexanamide (250 mg), catalyst (25 mg, 0.5 wt% of metal) and H₂ (50 bar) at 325 °C for 5 h), *N*-hexylhexanamide hydrogenolysis using Ru/C resulted in a significantly reduced methane yield (12%) compared to Ru/CeO₂ (72%), which is less than the CeO₂ support alone (24%, Supplementary Table 4). It is intriguing that the smaller size (2 nm) and more even distribution of the Ru NPs in Ru/C (Supplementary Fig. 10) does not translate to higher C–C hydrogenolysis activity compared to Ru/CeO₂. This disparity in activity might be due to the presence of Lewis acidic sites adjacent to the Ru NPs in the Ru/CeO₂ catalyst[54]. These sites create synergistic interactions with the Ru active sites, enhancing the overall catalytic performance. Moreover, CeO₂ permits rapid desorption of the formed ammonia, which prevents it from binding to the noble metal sites[55,56]. To further explore the effect of the support, catalysts comprising Ru NPs supported on other oxides (Al₂O₃, SiO₂ and TiO₂) were prepared via the same incipient wetness impregnation method used for Ru/CeO₂. The hydrogenolysis activity of Ru supported on CeO₂ remained the highest among all the Ru catalysts (Supplementary Fig. 11), possibly due to a higher tolerance to water and ammonia formed during the reaction[49,56]. A similar trend was observed for the Pt-based catalysts, where a lower *n*-hexane yield of 36% was obtained with Pt/C compared to an 82% yield for the Pt/CeO₂ catalyst. The product selectivity of the carbon-supported catalysts approached those of the ceria-supported catalysts when the reaction time was extended to 24 h, indicating that the selectivity for C–C bond hydrogenolysis is an intrinsic property of the metal NPs.

## Influence and optimisation of the reaction conditions

The catalytic activity of Ru/CeO₂ and Pt/CeO₂ in the hydrogenolysis of *N*-hexylhexanamide is strongly impacted by the hydrogen pressure and reaction temperature. As expected, too low a hydrogen pressure reduces the reaction rate, but too high a pressure can saturate the catalyst surface active sites[57]. From our reaction set-up, the optimal hydrogen pressure was found to be 50 bar, as reactions performed at 25 and 75 bar resulted in lower conversion rates for both catalysts (Supplementary Fig. 12). Increasing the reaction temperature from 325 to 350 °C was observed to result in a corresponding increase in both the overall conversion and yield of the major product for both catalysts (Fig. 3a), as higher temperatures facilitate heat and mass transfer to increase reaction rate[58]. This is particularly important for heterogeneous catalysis of high viscosity substrates, such as plastic waste, where efficient mass transfer is often critical[59].

It is noteworthy that the effect of reaction temperature appears to have a significant impact on C–C hydrogenolysis using the Ru/CeO₂ catalyst. At lower reaction temperatures, such as 250 °C, only C–O and C–N bond cleavage occurs and the major product is shifted towards higher-order alkanes, while at higher temperatures, such as 350 °C, the reaction proceeds with enhanced activity and selectivity towards methane, obtained in a maximum yield of 98%. In contrast, with the Pt/CeO₂ catalyst the reaction temperature has a more significant impact on C–O and C–N hydrogenolysis. At 250 °C, only 24% conversion was achieved, whereas at 325 °C, a significant improvement was observed, reaching a conversion of 99% and a maximum yield of *n*-hexane of 89%. Higher temperatures also promote C–C hydrogenolysis, such that a slight decrease in *n*-hexane yield is accompanied by a slight increase in yields of methane and *n*-pentane at a temperature of 350 °C.

The kinetics of *N*-hexylhexanamide conversion were investigated at 325 °C (Fig. 3b), demonstrating that Ru/CeO₂ exhibits faster conversion rates compared to Pt/CeO₂, indicating that the Ru NPs are more effective in the initial C–O bond cleavage. The Ru/CeO₂ catalyst achieved 95% conversion of the amide into alkane products within the first hour, indicating its strong hydrogenolysis activity[50]. In contrast, Pt/CeO₂ only achieved 34% overall conversion after 1 h, with a significant proportion of non-alkane hydrocarbons such as dihexylamine (Supplementary Table 7). This difference suggests that C–N hydrogenolysis is also slower for Pt/CeO₂ compared to Ru/CeO₂, in agreement with a previous study[60], resulting in the observation of intermediate amines. However, the selectivity towards *n*-hexane was still higher than dihexylamine, with a yield of 19% and 6%, respectively, after 1 h.

## Mechanistic inferences of product selectivity

Previous theoretical investigations at comparable reaction conditions have shown that Ru(0001) requires similar energy to perform C–C bond cleavage as the initial C – H bond activation required to remove hydrogen atoms from the chain end position[31], To provide new insights into the distinct catalytic behaviour of Pt NPs and Ru NPs, density functional theory (DFT) calculations were performed to study possible bond cleavage scenarios on the Pt(111) surface. The reduced metal Pt(0) form is the major surface species under reductive conditions, and has been reported to be the active species for hydrogenolysis (Fig. 4)[61,62]. Propane (C₃) was used as the model substrate, as the C–O and C–N bonds are typically cleaved prior to the C–C bonds (i.e. alcohol products and amines with C₍ₙ₌₆₎ were not observed as major

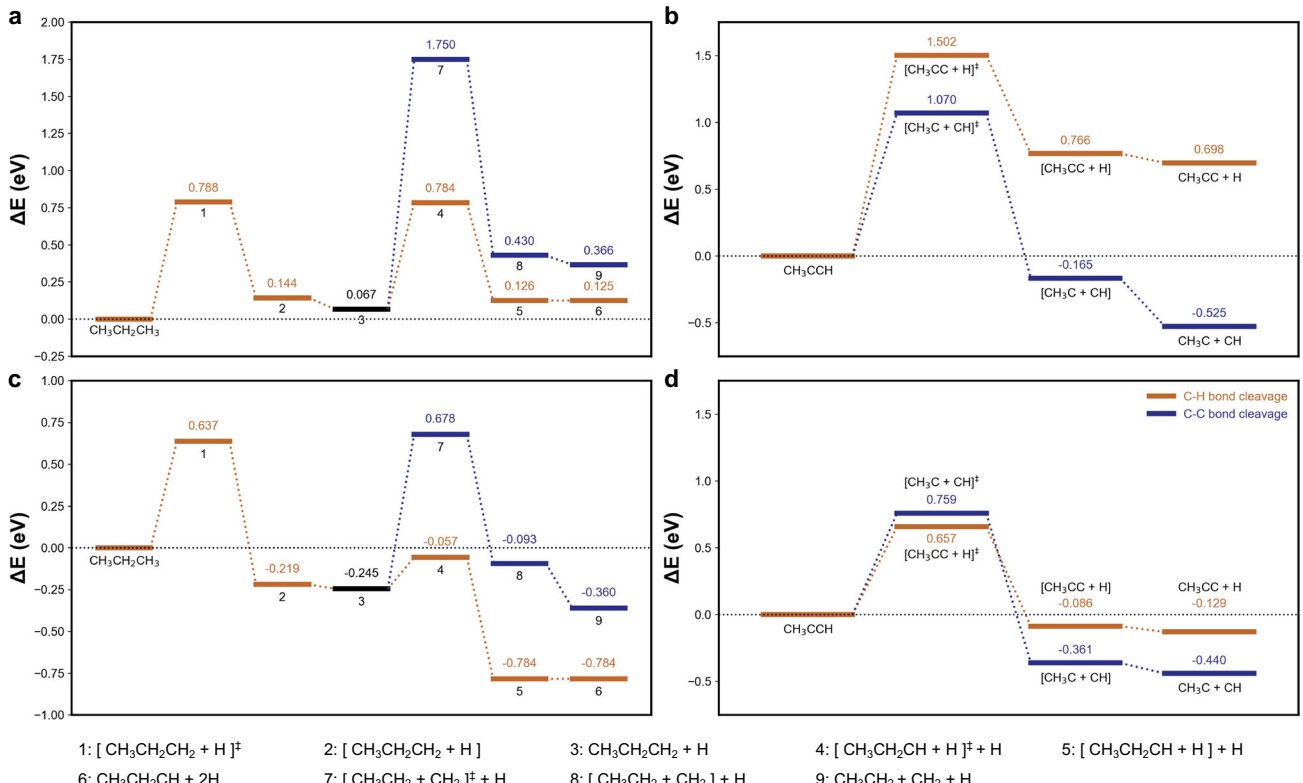

**Fig. 4 | DFT energy profiles for C−C bond and C−H bond hydrogenolysis on Pt(111) and Ru(0001) surface. a**, **b** Energy profiles for (**a**) the initial propane chemisorption, and (**b**) low H-content conditions, calculated on the Pt(111) surface. **c**, **d** Energy profiles for (**c**) the initial propane chemisorption, and (**d**) low H-content conditions, calculated on the Ru(0001) surface. Adapted from ref. 31, with permission from Elsevier. [] represents the as-dissociated species; []$^{‡}$ represents the transition states; see Supplementary Fig. 14 and Table 9.

products) and accordingly exhibit limited influence towards the selectivity of hydrocarbon products. At the initial stage of the reaction (i.e. high-H content species; scenario I in Supplementary Table 8 and Fig. 13), C−C cleavage on the Pt(111) surface requires a significantly higher activation energy than C−H cleavage ($\Delta E_a$ = 1.683 vs. 0.717 eV). In addition, C−H cleavage is an exothermic process on the Ru(0001) surface (−0.245 eV) upon the formation of the CH$_3$CH$_2$CH$_2$ intermediate, whereas it is an endothermic process on the Pt(111) surface (+ 0.067 eV). This indicates that C−C bond cleavage is less likely to occur at high-H content scenario and the formation of the key intermediates is relatively slower on Pt(111) surface. The C$_3$ species may also undergo several C−H cleavages, to generate the low-H content intermediate CH$_3$CCH, in which both the $\alpha$-C and $\beta$-C atoms are adsorbed on the metal surface (Scenario II)[31,63]. At this stage, the activation energy for C−C cleavage becomes lower compared to high-H content species on the Pt(111) surface ($\Delta E_a$ = 1.070 vs. 1.683 eV). However, it still requires higher energy than the initial C−H bond cleavage ($\Delta E_a$ = 1.070 vs. 0.788 eV). The computational results indicate that the Pt surface is less active both thermodynamically and kinetically in cleaving both C−C and C−H bonds compared to the Ru surface, hence Pt/CeO$_2$ tends to preserve the carbon backbone after hydrogenolysis of the amide bond.

## Conversion of pure PAs, a commercial PA product, PA-blends and PA-composites

Having optimised the conditions (325 °C, 50 bar H$_2$) for the hydrogenolysis of *N*-hexylhexanamide, the depolymerisation of four common PA resins, i.e. PA-6, PA-66, PA-12 and PA-612 (Supplementary Figs. 14 and 15), was evaluated at a longer reaction time of 24 h (Fig. 5). The longer reaction time is required to achieve high conversion due to the higher melt-viscosity of these resins, which greatly slows down the

rate of hydrogenolysis. For instance, with Ru/CeO$_2$, the conversion of PA-6 was lower when the experiment was performed without stirring (96% with stirring vs. 78% without, Supplementary Table 9, entry 9). The Ru/CeO$_2$ catalyst leads to near quantitative conversion for all PA resins, with methane being the primary hydrocarbon product in all cases, obtained in up to 99% yield for PA-12 and PA-612. This provides a power-to-methane process that can potentially be directly utilised by existing natural gas networks[31]. In contrast, the Pt/CeO$_2$ catalyst results in slightly lower conversion rates, with 88% conversion for PA-6, 78% for PA-66, 98% for PA-12 and 99% for PA-612. Similar to the previous studies on model amides, Pt/CeO$_2$ also showed a preference for the formation of higher-order alkanes, such as *n*-hexane or *n*-dodecane, depending on the length of the carbon backbone in the polymer. These longer chain hydrocarbons have uses as non-polar solvents with multiple applications including in printing, adhesives, textile manufacture etc., and in the case of *n*-dodecane, as a jet fuel[64]. This enables selectivity in the length of the alkane products, depending on the metal species of the CeO$_2$ supported catalyst. Ammonia was detected for all conversions, and the nitrogen atoms in the PAs are expected to be hydrogenated into ammonia once the C−N bonds are cleaved.

Overall, both catalysts show excellent yields and major product selectivities in converting pure PA resins into alkanes, predominantly methane for Ru/CeO$_2$ and *n*-hexane and/or *n*-dodecane for Pt/CeO$_2$. The differences in overall conversion observed may be attributed to the varying properties of the different PA resins. Lower specific heat capacities in PA-12 and PA-612 result in lower melt viscosities, facilitating better mass and heat transfer, which contributes to a higher overall conversion[65,66]. The Ru/CeO$_2$ catalyst is able to fully convert PA-6 in just 2 h with an 86% yield of methane (Supplementary Table 11), which is remarkable as it represents a faster conversion rate for the polymer than that observed for N-hexylhexanamide. This highlights

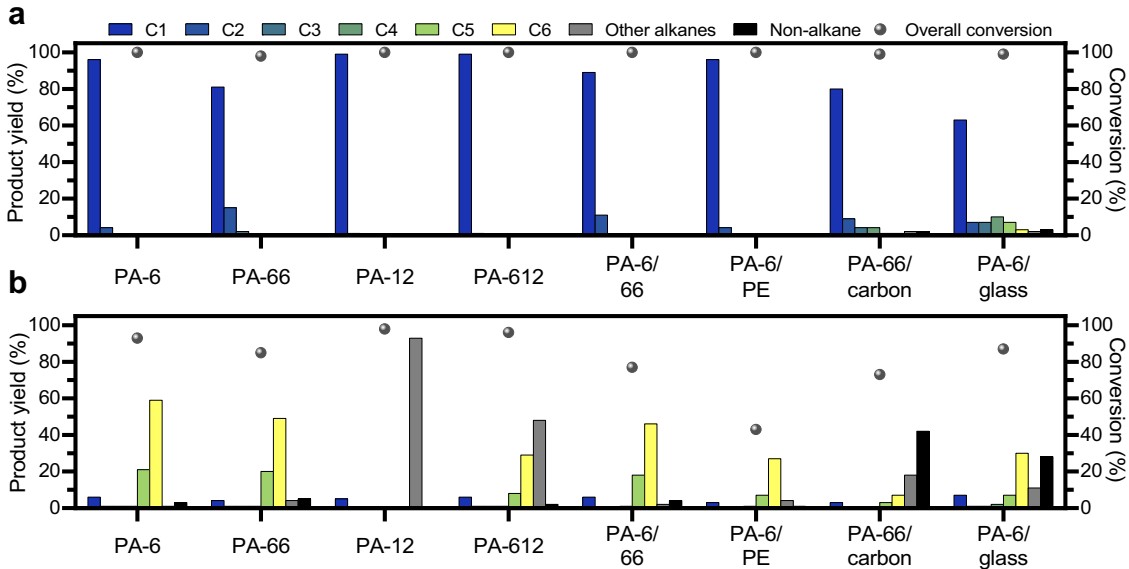

**Fig. 5 | Hydrogenolysis of pure PAs, PA-blends, and PA-composites catalysed by M/CeO₂ (M = Ru, and Pt). a** Catalysed by Ru/CeO₂ and **b** catalysed by Pt/CeO₂. Reaction conditions: polymer (250 mg), catalyst (25 mg, 0.5 wt% of metal) and H (50 bar) at 325 °C for 24 h. PA-66/carbon and PA-6/glass composites contain 30 wt% of the composite material, which were not included in the conversion calculation. The PA-6/PE blend was a 1:1 mixture (125 mg:125 mg) of their pure polymers. Supplementary Table 9 provides detailed product yields.

the influence of both activation enthalpies and entropies on the rate of hydrogenolysis[67]. The conversion of PA-6 with Pt/CeO₂ achieved a maximum *n*-hexane yield of 64% when reacted for 18 h (Supplementary Table 11, entry 4). A longer reaction time led to a higher occurrence of C−C hydrogenolysis, resulting in a decrease in the selectivity of *n*-hexane, accompanied by a proportional increase in the yield of methane and *n*-pentane. However, further reduction of reaction time to 12 h resulted in a decrease in *n*-hexane yield to 34%, as the overall conversion became much lower (Supplementary Table 11, entry 3). Following these results, the catalysts were tested on the conversion of a commercial PA-66 fishing net (Supplementary Fig. 16) with 2.6 g of the fishing net converted into 1.4 g of methane and 0.15 g of ammonia by the Ru/CeO₂ catalyst and 1.8 g of hexane and 0.14 g of ammonia by the Pt/CeO₂ catalyst.

Encouraged by the effectiveness of the catalysts in PA hydrogenolysis, depolymerisation of Ultramid® C40L, an industrial-grade blend of PA-6 and PA-66 (PA-6/66), was evaluated. This method provides a novel approach to the selective conversion of a PA-6/66 blend into water, ammonia and alkanes, with the Ru/CeO₂ catalyst leading to a near-quantitative yield of methane (98%) and the Pt/CeO₂ catalyst achieving a conversion of 73% with *n*-hexane obtained in 63% yield. This outcome is particularly noteworthy as it highlights the effectiveness of the Pt/CeO₂ catalyst in retaining the alkyl-backbone of both PA-6 and PA-66. Beside the PA-based blend, the hydrogenolysis of PA-6 blended with polyethylene (PA-6/PE) was also studied. Due to the difference in C−C hydrogenolysis ability between the two catalysts, Ru/CeO₂ was able to fully convert the PA-6/PE blend into methane (96% yield), whereas Pt/CeO₂ was able to convert only the PA fraction of the blend evidenced by *n*-hexane as the major product (27% yield) with 46% overall conversion. The ability of Pt/CeO₂ to give a single hydrocarbon product from PA blends is highly advantageous compared to routes that give mixtures of monomers, which would then require challenging separation. The same is true for Ru/CeO₂, which has the added advantage of being able to convert PA/polyolefin blends into methane as the only hydrocarbon product.

The hydrogenolysis of PA-based composites was also investigated to assess the performance of the catalysts in the presence of other widely used composites materials. Despite the large proportion (30 wt.%) of composites such as carbon or glass, the Ru/CeO₂ catalyst is able to achieve reasonably high methane yields (89% for PA-66/carbon and 61% for PA-6/glass) relative to the mass of PA present in the composite, indicating that the catalytic performance is not significantly affected by the presence of composites, which highlights the robustness of the catalyst. The conversion of PA composites catalysed by Pt/CeO₂ requires a longer reaction time of 72 h. The yield of *n*-hexane reached 40% and 52% for the conversion of PA-66/carbon and PA-6/glass, respectively. The rate of hydrogenolysis is slower with composite materials compared to pure PA, presumably because the carbon and glass hinder access to active metal surface sites.

### Catalyst recycling

To further understand their applicability towards industrial processes, the catalysts were tested for their reusability (Fig. 6). The robustness of the catalysts was demonstrated through repeated hydrogenolysis cycles using PA-6 for Ru/CeO₂ and PA-12 for Pt/CeO₂. Even after five reaction cycles, full conversion was still attainable with minimal impact on product selectivity for both catalysts. Furthermore, for Pt/CeO₂, the *n*-dodecane selectivity improved after the first cycle, from 60% yield to 76% yield in the second cycle. In the fifth cycle, the yield of *n*-dodecane reached a maximum of 79%, which demonstrates that the C−C bond cleaving ability of Pt/CeO₂ decreases after each reaction. STEM images of the used catalysts revealed little to no change in the size or distribution of the Ru and Pt metal NPs after hydrogenolysis. The M-O-Ce bond acts as an anchor and prevents the metal NPs from agglomerating at 325 °C[48]. The lack of sintering emphasises the robustness and potential for long-term use of these catalysts.

### Discussion

Catalytic hydrogenolysis of PAs may be performed efficiently by M/CeO₂ (M = Ru, Rh, Pd, Ir, or Pt) with Ru/CeO₂ and Pt/CeO₂ also displaying excellent selectivities. The M/CeO₂ catalysts show different behaviours in hydrogenolysis of the amide bond, due to having different intrinsic barriers to C−C and C−N bond cleavage. Ru/CeO₂ was found to be the most active in catalysing hydrogenolysis of PA materials rapidly into water, methane and ammonia, whereas Pt/CeO₂ is less active both thermodynamically and kinetically in cleaving both the C−C and C−H bonds, allowing a selectivity to higher-ordered hydrocarbons, notably *n*-hexane.

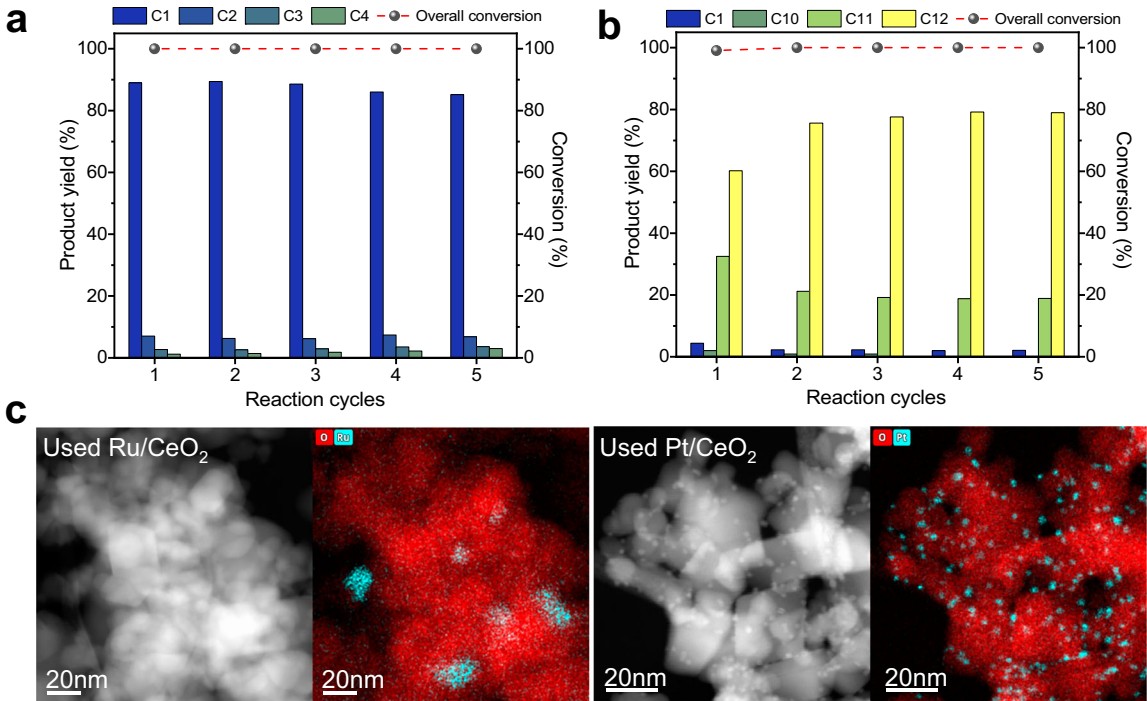

**Fig. 6 | Recycling experiments of Ru/CeO₂ and Pt/CeO₂ catalysts. a** Product distribution of PA-6 conversion with Ru/CeO₂ over five cycles. **b** Product distribution of PA-12 conversion with Pt/CeO₂ over five cycles. Reaction conditions: polymer (250 mg), catalyst (25 mg, 0.5 wt% of metal) and H₂ (50 bar) at 325 °C for 2 hours (Ru/CeO₂) and 24 h (Pt/CeO₂). **c** STEM-HAADF (left-hand side) and EDX images (right-hand side) of the used catalysts. Supplementary Tables 12 and 13 provides detailed product yields.

The Ru/CeO₂ and Pt/CeO₂ catalysts are able to transform PA composites or blends and a PA-66 fishing net into high yields of methane or higher-order hydrocarbons, together with ammonia. Beyond the high selectivity the Ru/CeO₂ and Pt/CeO₂ catalysts, they are easy to prepare, highly robust, as evidenced by the stability of their conversion and product selectivity over multiple reaction cycles and allow facile separation and reuse. Moreover, solvent is not required as the reaction is performed above the melting point of the PA substrate with solvent being formed as the reaction progress, which possibly contributes to the stability of the catalysts by preventing coking. The conversion of a PA-66 fishing net was conducted on the gram scale, demonstrating the potential for industrial scale-up applications. These findings provide a foundation for designing future catalysts for the conversion of various types of PA waste, including PA composites and blends, which are commonly found in PA-containing products[28–30]. In addition, the ability to selectively convert mixtures of PA and polyolefin waste into two easily separated products, i.e., methane and ammonia, is of immense potential value, as the separation of waste plastics is challenging. Based on the utility and value of the products obtained from PAs by these catalysts, the incentive to collect PA waste, especially in the ocean, should increase.

## Methods
### Materials
All chemicals were used as purchased without further purification. RuCl₃.3H₂O, RhCl₃.3H₂O, PdCl₂, IrCl₃.3H₂O and K₂PtCl₄ were purchased from PreciousMetalsOnline. CeO₂, hexylamine, hexanoyl chloride and 5% Ru/C were purchased from Sigma-Aldrich. 5% Pt/C was purchased from Alfa Aesar. *p*-Xylene was purchased from Acros Organics. PE (powder), PA-6 (resins, $M_n$ = 27,380 g/mol), PA-66 (resins, $M_n$ = 47,240 g/mol), PA-12 (resins, $M_n$ = 27,060 g/mol), PA-612 (resins, $M_n$ = 14,730 g/mol), 30% carbon-fibre PA-66 composite (resins, $M_n$ = 19,920 g/mol) and 30% glass-fibre reinforced PA-6 rods were purchased from Sigma-Aldrich. The rods were sawn into smaller pieces

before reaction. Ultramid® C40 L (resins, $M_n$ = 49,010 g/mol) was provided by BASF. Gel permeation chromatography was used to measure the $M_n$ of all resins, except for PE and the PA composites. PA-66 fishing net was purchased from Stucki Thun.

**Synthesis of the M/CeO₂ catalysts.** A modified two-step annealing procedure was used to obtain ultra-high-density atomic clusters deposited on CeO₂[46]. K₂PtCl₄ (0.213 g) and CeO₂ nanopowder (1 g, <25 nm, BET) were dispersed in ultrapure water (20 mL) and sonicated for 10 min, and the water was then removed on a rotary evaporator. The resulting powder was calcined at 300 °C in a tube furnace at a heating rate of 5 °C/min, for 5 h under a flow of argon (200 mL/min). After cooling to room temperature, the powder was washed with ultrapure water (2 × 15 mL), then filtered and dried under vacuum. The powder was then calcined in a tube furnace at 550 °C, at a heating rate of 1 °C/min, then for 5 h under static air to produce Pt/CeO₂. A similar procedure using RuCl₃.3H₂O (0.259 g), RhCl₃.3H₂O (0.256 g), PdCl₂ (0.167 g) and IrCl₃.3H₂O (0.184 g) was performed for the synthesis of Ru/CeO₂, Rh/CeO₂, Pd/CeO₂ and Ir/CeO₂, respectively.

**Characterisation of M/CeO₂ catalysts.** Powder X-ray diffraction (PXRD) of all M/CeO₂ catalysts was measured from 2θ = 20–80° on a Bruker D8 Discover Vario diffractometer by using Cu Kα (L = 1.54 Å) radiation that was operated at 40 kV and 40 mA. X-ray photoelectron spectroscopy (XPS) measurements were carried out on an Axis Supra (Kratos Analytical) using the monochromated Kα X-ray line of an Aluminium anode. The pass energy was set to 40 eV with a step size of 0.15 eV. The samples were electrically insulated from the sample holder and an electron flood gun was used to neutralise the charges. The binding energy scale was then referenced at 284.8 eV using the aliphatic line of the C1s orbital. Inductively coupled plasma mass spectrometry (ICP-MS) was performed on a Perkin Elmer NexIon 350 spectrometer for elemental analysis. Scanning transmission electron microscopy (STEM) images for Rh/CeO₂, Pd/CeO₂ and Ir/CeO₂

were captured on a FEI Tecnai Osiris electron microscope at 200 KeV, while high resolution (HR) STEM images for Ru/CeO$_2$ and Pt/CeO$_2$ were captured on a FEI Titan Themis electron microscope at 300 keV. High-angle annular dark field (HAADF) imaging mode was used for all STEM imaging. Energy-dispersive X-ray spectroscopy (EDX) was performed at 20 KeV.

**Synthesis of model compound N-hexylhexanamide.** N-hexylhexanamide was synthesised using a literature procedure with slight modifications[68]. A solution of sodium hydroxide (3 g, 75 mmol) in water (30 mL) was prepared and hexylamine (3.03 g, 30 mmol) was added, followed by stirring for 5 min. Next, hexanoyl chloride (3.37 g, 25 mmol) was added dropwise under vigorous stirring, and the resulting mixture was stirred for 3 h. The product was extracted with dichloromethane (15 mL), and the organic layer was washed with water (3 × 15 mL) and dried with sodium sulfate. The dichloromethane was removed under reduced pressure to yield the solid N-hexylhexanamide compound. (see Supplementary Fig. 4 for $^1$H NMR spectrum) $^1$H NMR (400 MHz, CDCl$_3$) δ 5.54 (s, 1H), 3.24−3.19 (q, 2H), 2.15−2.11 (t, 2H), 1.65−1.57 (m, 2H), 1.50−1.43 (m, 2H), 1.31−1.25 (m, 10H), 0.89−0.85 (m, 6H).

**Reaction set-up and product analysis for the conversion of N-hexylhexanamide.** In a typical reaction, M/CeO$_2$ catalyst (25 mg nominal, 0.5 wt% of metal with respect to the substrate) and N-hexylhexanamide (250 mg) were added to a 20 mL glass vial with a glass-coated magnetic stir-bar. The vial was then placed inside a 68 mL Parr autoclave, which underwent three cycles of hydrogen (H$_2$) cycling and was pressurised to a target pressure of 50 bar. The autoclave was placed in a pre-heated (250−350 °C) Parr ceramic heater, and stirring was maintained at 700 r.p.m. To stop the reaction, the autoclave was removed from the heater and placed under running water to cool to room temperature.

The gaseous products generated during the reaction were extracted in a gas-sampling bag and analysed using gas chromatography with a flame ionization detector (GC-FID). Ammonia was quantified using a separate quantification method as described in the quantification of ammonia section. The liquid products remaining in the autoclave were extracted with diethyl ether, separated from the solid catalyst by filtration, and analysed using an Agilent 7000 C GC-MS instrument with p-xylene as the internal standard. The overall conversion of N-hexylhexanamide was determined by subtracting the percentage of substrate remaining after the reaction from the initial amount of substrate used. The yield of each product was calculated from the carbon balance, using the following equation:

$$\text{Yield of product} = \frac{(\text{mol of carbon in product})}{(\text{mol of carbon in starting substrate})} \times 100\%$$

**Conversion of real polymers.** The procedure for polymer hydrogenolysis was similar to that for N-hexylhexanamide, with the exception that diethylether was used to extract liquid products and the solid residue was weighed after drying. The overall conversion was calculated from the mass difference at the end of the reaction, and solid products were not included in the conversion calculation.

**Ammonia detection and quantification.** To test for the presence of ammonia in the products, a 5 mL sample of the gaseous product was extracted with a gas-tight syringe at ambient pressure and slowly bubbled into 10 mL of miliQ water. Calibration standards for ammonia were prepared by bubbling different volumes of pure ammonia gas into 10 mL of miliQ water. The indophenol reagent was prepared according to a previously reported method[69]. After adding the indophenol reagent to the sample or standard, the solution was allowed to stand in the dark for 1 h before measuring the absorbance at 651 nm

using a PerkinElmer Lambda 850 + UV-Vis spectrophotometer. The absorbance value of the characteristic maximum peak of the indophenol blue complex was recorded and compared with the calibration standards.

**Recycling experiments.** To evaluate the robustness of the catalysts, their performance was studied in repeated cycles of hydrogenolysis of PA-6 (for Ru/CeO$_2$) or PA-12 (for Pt/CeO$_2$). The reaction was carried out using 250 mg of PA-6 (or PA-12) and Ru/CeO$_2$ (or Pt/CeO$_2$) (25 mg, 0.5 wt% of metal), with a hydrogen pressure of 50 bar at 325 °C for 2 h (or 24 h for the Pt-catalysed reaction). After completion of each reaction, the catalyst was washed with diethyl ether and left to dry in an oven overnight at 80 °C. A fresh batch of PA-6 or PA-12 (250 mg) was added to the reaction vessel with the dried catalyst for the next cycle and this procedure was repeated for a total of five cycles.

**Computational details.** All computations were performed using density function theory (DFT) with the Perdew−Burke−Ernzerhof (PBE) functional[70], as implemented in the Vienna Ab-initio Simulation Package (VASP)[71]. The valence electron wave functions were expanded in plane-wave basis sets with a 400 eV cutoff and the projector augmented wave (PAW) method was used to describe the core-electron interactions[72]. A 6 × 6 × 1 k-point grid was sampled using the Monkhorst−Pack scheme for each computation. All structures were fully optimised until the residual forces on the constituent atoms became less than 0.02 eV Å$^{-1}$, while keeping the lattice parameter fixed, which was taken from the experiment. For transition-state searching, the climbing image nudged elastic band (cNEB) method was performed together with the dimer approach[73−75]. The frequency computation was performed using finite difference approach (POTIM = 0.02) for every optimised structure of the transition state to ensure only one imaginary mode was present along the reaction coordinate.

**Structural construction.** The lattice parameters of the primitive FCC Pt crystal were first optimised ($a$ = 2.775 Å), and a 3 × 3 4-layer (111) slab with a 15 Å vacuum layer was then created. The conformational search for each C$_3$ species was then performed to find the adsorbate configuration with the lowest energy. The energies of these conformers and the transition states were then used to construct the reaction energy diagram.

## Data availability
Experimental data that support the findings of this study have been deposited in Zenodo https://doi.org/10.5281/zenodo.8356468.

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

## Acknowledgements

This publication was supported by the EPFL (Switzerland) and NCCR catalysis (grant number 180544), a National Centre of Competence in Research funded by the Swiss National Science Foundation (SNSF). The authors thank the National Center for High-performance Computing (NCHC), Taiwan, for providing computational and storage resources. We thank Daniel Ortiz, Francisco Sepulveda, Natalia Gasilova, Mensi Mounir, Pascal Schouwink and Victor Boureau for their technical support.

## Author contributions

X.B.W., W.-T.L., R.C.T.-R. and P.J.D. contributed to the design of the experiments and data analysis. X.B.W. and P.C.L.D. performed the experiments and L.K.-H. performed the DFT calculations; X.B.W. and P.J.D. wrote the manuscript and all authors discussed, commented on and proofread the manuscript.

## Competing interests

The catalytic method is described in a patent invented by X.B.W., W.-T.L. and P.J.D (EP Patent, No. 2315963.1). The remaining authors declare no competing interests.
