## [Peer Review File · Nature Communications]

REVIEWER COMMENTS

Reviewer #1 (Remarks to the Author):

In this manuscript, the hydrogenation of amides and polyamides to hydrocarbons as methane, hexane or dodecane using different precious metal based heterogeneous catalysts is presented. The recycling of polymer waste is a topic of high current interest to address a global challenge. Therefore, new approaches to utilize plastic waste are highly welcome. From a catalysis point of view (synthesis, characterization, reactivity, recyclability), the work seems to be solid. Nevertheless, there are certain concerns about the usability of this approach and how the state of the art is taken into account. Therefore acceptance for nature communications is not recommended.

Introduction lines 57-60: Some work is missing on polyamide recycling via hydrogenative depolymerization in presence of NH₃ which should be cited and taken into account:

GB1179706 and DE1695282A1 (hydrogenation of polyamides in the presence of NH₃ using Ru- and Ni catalysts)

ACS Sustainable Chem. Eng. 2022, 10, 3048-3056 (hydrogenation of polyamides in the presence of NH₃ using Nb₂O₃ and a Ru catalyst).

Line 60-63: In the before mentioned paper, also polyamides containing up to 30% carbon black was depolymerized. Please take into account.

Line 85-87 and lines 191-195. See the supporting information page tables S3 and scheme S3 of ref 20. There hydrogenative C-heteroatom cleavage in polyamide was observed with a heterogeneous Ru catalyst and hexane formed.

Figure 2, figure 3, figure 5 and figure 6: The authors should use other colors for the columns. It is very difficult to get, what is which column.

Generally, and considering the other work on polyamide depolymerization: Here the authors are making methane, hexane or dodecane. In other work, one gets at least functional molecules back as amines or alcohols. With methane or hexane, one cannot do that many things besides just burning it or making

syngas out of it (and then go into the syngas value chain). But why then not just burn the polyamide to get the heating value or making syngas rather than adding a high-pressure hydrogenation step before with precious metal catalysts and losing all functionalities? The authors must point more out, where is the significant advantage of this approach compared to others, where one would get more functional small molecules back.

Supporting information, Table 1:

The old work on the heterogeneous hydrogenation of polyamides with Ru- and Ni catalysts in the presence of ammonia (GB1179706 and DE1695282A1; both patents accessible via espacenet from the EPA) should also be taken into account here.

The recent work in ACS Sustainable Chem. Eng. 2022, 10, 3048-3056 should also be taken into account here.

The entry with the hydrogenation (homogeneous) is not correct. Ref 6 is cited, but in ref 6, polyamides were hydrogenated in DMSO, not THF. And in ref 6 also PA66 is hydrogenated to the diol and diamine, see entry 4 in table 2 in this ref. Not only 6-amino-1-hexanol. THF was used in the system presented in ref 20 of the manuscript. By the way, in ref 20, the products obtained are 1,6-hexamethylenediamine and 1,6-hexandiol in yields up to 78% for a model PA (see table 2 in ref 20). This should also be reflected here.

Reviewer #2 (Remarks to the Author):

The manuscript titled "Controlling the selectivity of the hydrogenolysis of polyamides catalysed by ceria supported metal nanoparticles" reports the selective hydrogenolysis of polyamide models and difficult-to-recycle 'real-world' polyamide waste using M/CeO₂ catalysts with high activities. Density functional theory has been used to rationalize selectivity, and the effect of hydrogenolysis conditions on catalyst activity and selectivity have been investigated. Overall, this is a very thorough, logical, and well-conducted study in an important field of research. Therefore, I recommend publication in Nature Communications after some minor corrections and suggestions detailed below:

1. The resolution of the TOC and images throughout the manuscript and supplementary information could be improved.
2. Introduction: please define PA-6 as Nylon-6 upon first use.

3. The catalyst characterization and theoretical studies relate very nicely to the experimental data, with detailed explanations provided. Have any attempts been made to look at the acidity of the catalysts? Differences in acidity could potentially explain the differences in activity observed for the Ru/MxOy catalysts. Additionally, H₂ chemisorption could also be used to quantify the surface area and dispersion of the metal NPs on the surface. Note: these are suggestions, not requirements for publication.

4. The catalyst screening section states: "A control reaction was performed to determine the thermal stability of N-hexylhexanamide under the reaction conditions. The control reaction indicates that the substrate undergoes low levels (6%) of thermal decomposition, confirming that the observed hydrogenolysis activity was due to the activity of the metal catalysts." The authors should consider mentioning here that 24.4% conversion is observed with just the CeO₂ support, with the conversion increasing when metal NPs are on the surface. The current sentence implies that all the observed activity is from the metal NPs, whereas some activity may also come from the support.

5. Fig 2b. The scheme is missing H₂O as a product on the first arrow.

6. Catalyst screening: Reference Supplementary Figure 2c when discussing PdCl₂.

7. Figures S11 and S12: add the overall conversion to the plots (similar to Figures 2 and 3 in the manuscript).

8. It's great that these catalysts are able to break down real-world plastic waste, PA blends, and PA composites. It would be beneficial to mention some of the uses and value of the products (methane, n-hexane, n-dodecane etc.) in the conclusions.

Reviewer #3 (Remarks to the Author):

Revisions:

The manuscript demonstrates the hydrogenolysis of various PAs catalysed by platinum-group metal nanoparticles (NPs) supported on ceria (CeO₂). Ru/CeO₂ was found to be the most active in catalysing hydrogenolysis of PA materials rapidly into water, methane and ammonia, whereas Pt/CeO₂ yields hydrocarbons retaining the carbon backbone of the parent PA. The DFT computations was also performed to rationalize the observed selectivity. The work is meaningful, and can be accepted after a major revision. My detailed comments are as follows:

In page 4, the authors mentioned that "there have been limited studies exploring the cleavage of carbon-heteroatom bonds in PAs". However, there are numerous experimental and theoretical studies have been reported about the homogeneous hydrogenolysis of PA and polyamides. Some literatures

(e.g. Chem. Sci., 2021, 12, 10590; JACS Au 2021, 1, 517–524; Inorg. Chem. 2022, 61, 14662–14672) should be introduced. What is more, the advantages of Ru/CeO₂ and Pt/CeO₂ over the homogeneous catalysts must be described in detail.

In Fig. 2, the N-hexylhexanamide undergoes the C-O hydrogenolysis to form a secondary amine intermediate, and the C-N hydrogenolysis to form a N-terminal amine. However, the amides are widely reported to undergo the hydrogenolysis to provide N-terminal amines and alcohols (ACS Catal. 2020, 10, 5511–5515; J. Am. Chem. Soc. 2019, 141, 12962–12966; Nat. Commun. 2015, 6, 6859), and the secondary amine intermediates are not involved. The mechanism in Fig. 2 could not be right.

The discussion of chemoselectivity is based on the reactions of CH₃CCH in scenario II of Fig. 4, but the theoretical study does not provide the pathway for the formation of CH₃CCH intermediate. Otherwise, the experiment should prove the formation of CH₃CCH.

Reviewer #1 (Remarks to the Author):

In this manuscript, the hydrogenation of amides and polyamides to hydrocarbons as methane, hexane or dodecane using different precious metal based heterogeneous catalysts is presented. The recycling of polymer waste is a topic of high current interest to address a global challenge. Therefore, ne approaches to utilize plastic waste are highly welcome. From a catalysis point of view (synthesis, characterization, reactivity, recyclability), the work seems to be solid. Nevertheless, there are certain concerns about the usability of this approach and how the state of the art is taken into account. Therefore acceptance for nature communications is not recommended.

We thank the reviewer for their positive appraisal of the work conducted and also their concerns which are commented on further below.

1. Introduction lines 57-60: Some work is missing on polyamide recycling via hydrogenative depolymerization in presence of NH₃ which should be cited and taken into account: GB1179706 and DE1695282A1 (hydrogenation of polyamides in the presence of NH₃ using Ru- and Ni catalysts). ACS Sustainable Chem. Eng. 2022, 10, 3048-3056 (hydrogenation of polyamides in the presence of NH₃ using Nb₂O₃ and a Ru catalyst).

We thank the reviewer for drawing our attention to these reports of hydrogenative ammonolysis, and we have added these references to the introduction (References 21-23).

Techniques explored include pyrolysis^{12,13}, hydrolysis^{14,15}, acid- or base-catalysed depolymerisation^{16,17}, ionic-liquid catalysed solvolysis¹⁸, metal-complex catalysed hydrogenation^{19,20}, and hydrogenative ammonolysis²¹⁻²³.

However, hydrogenative ammonolysis is fundamentally different to our approach, the former uses an excess of ammonia, whereas the approach we report generates ammonia, which can be easily separated from the hydrocarbon products. Moreover, in the present work, mixed plastic waste (containing polyamides, polyolefins, and structural additives) may be transformed, which excludes the need for the challenging task of plastic waste separation. Overall, we believe that all these methods are important and are potentially viable in different situations.

2. Line 60-63: In the before mentioned paper, also polyamides containing up to 30% carbon black was depolymerized. Please take into account.

As requested, we have accounted for this in both the introduction and Supplementary Table 1:

Two of these studies investigated the conversion of PA-blends (containing a mixture of different PAs) or PA composites^{13,21}, which contain PA blended with other materials, such as carbon fibre or glass.

Supplementary Table. 1 | Current routes for the chemical recycling of polyamides.

Process	Conditions	Substrate		Product (yield)	Ref.
		Pure PA	PA-composite/ blend		
Pyrolysis (Heterogenous)	410°C Calcined scallop shells High-purity He gas flow	PA-6	–	Caprolactam (66%)	1
Hydrothermal (Homogenous)	345°C 90 bar water	PA-6	–	Caprolactam (89%)	2
Solvolytic (Homogenous)	270°C Glycolic acid MeOH	PA-66	–	Adipic acid (75%)	3
Pyrolysis (Heterogenous)	360°C KOH/ α -Al ₂ O ₃	PA-6	–	Caprolactam (85%)	4
Solvolytic (Homogenous)	300°C Ionic acid DMAP	PA-6/ PA-12	–	Caprolactam (86%)/ Lauro lactam (7%)	5
Hydrogenation (Homogenous)	200°C Ru pincer complex and KO ^t Bu THF 100 bar H ₂	PA-66	–	Diamine (78%) Diol (62%)	6
Hydrogenative ammonolysis (Heterogenous)	200°C Nb ₂ O ₅ RuWO ₄ /MgAl ₂ O ₄ CPME 30 bar H ₂ 6 bar NH ₃	PA-1010/ PA-11/ PA-12	30% carbon	Primary amines (62%) Secondary amines (36%) Diamines (43%)	7
Hydrogenolysis (Heterogeneous)	325°C Ru/CeO ₂ or Pt/CeO ₂ 50 bar H ₂	Multiple Pas	30% carbon 30% glass	Methane and ammonia (>99%), or higher-order hydrocarbons and ammonia (93%)	This work

3. Line 85-87 and lines 191-195. See the supporting information page tables S3 and scheme S3 of ref 20. There hydrogenative C-heteroatom cleavage in in polyamide was observed with a heterogeneous Ru catalyst and hexane formed.

The heterogeneous catalysts used in ref 20 (*ChemSusChem* **2021**, *14* (19), 4176–4180) were employed as controls, used to confirm that their reaction system is homogenous. We have now commented on this work in the introduction:

Despite the various studies which focus on the design of heterogeneous catalysts for selective C–O bond hydrogenolysis,^{39–41} there have been limited studies exploring the cleavage of carbon–heteroatom bonds in PAs. However, a supported-Ru catalyst was used to confirm the homogeneity of a Ru-complex used to catalyse PA hydrogenation²⁰.

4. Figure 2, figure 3, figure 5 and figure 6: The authors should use other colors for the columns. It is very difficult to get, what is which column.

We have changed the colour scheme of all the bar columns following the reviewer's suggestion.

5. Generally, and considering the other work on polyamide depolymerization: Here the authors are making methane, hexane or dodecane. In other work, one gets at least functional molecules back as amines or alcohols. With methane or hexane, one cannot do that many things besides just burning it or making syngas out of it (and then go into the syngas value chain). But why then not just burn the polyamide to get the heating value or making syngas rather than adding a high-pressure hydrogenation step before with precious metal catalysts and losing all functionalities? The authors must point more out, where is the significant advantage of this approach compared to others, where one would get more functional small molecules back.

We have added further discussion to better emphasize the advantages of our approach, emphasised the formation of ammonia and the uses of the hydrocarbon products. The difference in C-C hydrogenolysis ability between Pt and Ru, which controls selectivity, is fundamentally interesting. Furthermore, the *n*-hexane and *n*-dodecane produced can be used as solvents. We have added a sentence to highlight the uses of the alkane products:

The Ru/CeO₂ catalyst leads to near quantitative conversion for all PA resins, with methane being the primary hydrocarbon product in all cases, obtained in up to 99% yield for PA-12 and PA-612. This provides a power-to-methane process which can potentially be directly utilized by existing natural gas networks³¹. In contrast, the Pt/CeO₂ catalyst results in slightly lower conversion rates, with 88% conversion for PA-6, 78% for PA-66, 98% for PA-12 and 99% for PA-612. Similar to the previous studies on model amides, Pt/CeO₂ also showed a preference for the formation of higher-order alkanes, such as *n*-hexane or *n*-dodecane, depending on the length of the carbon backbone in the polymer. These longer chain hydrocarbons have uses as non-polar solvents with multiple applications including in printing, adhesives, textile manufacture etc., and in the case of *n*-dodecane, as a jet fuel⁶⁵.

We have emphasized the formation of ammonia in the discussion:

The Ru/CeO₂ and Pt/CeO₂ catalysts are able to transform PA composites or blends and a PA-66 fishing net into high yields of methane or higher-order hydrocarbons, together with ammonia.

We have emphasised in the discussion the significant advantage of the M/CeO₂ catalysts which is its ability to valorise mixed plastic waste:

In addition, the ability to selectively convert mixtures of PA and polyolefin waste into two easily separated products, i.e., methane and ammonia, is of immense potential value, as the separation of waste plastics is challenging.

Also see response to reviewer 3, comment 2 where we specifically highlight the advantages of our heterogeneous catalysts.

6. Supporting information, Table 1:

The old work on the heterogeneous hydrogenation of polyamides with Ru- and Ni catalysts in the presence of ammonia (GB1179706 and DE1695282A1; both patents accessible via Espacenet from the EPA) should also be taken into account here.

The recent work in ACS Sustainable Chem. Eng. 2022, 10, 3048-3056 should also be taken into account here.

As described above, we have taken these studies into account in both the main text (References 21-23) and the SI (Reference 7).

7. The entry with the hydrogenation (homogeneous) is not correct. Ref 6 is cited, but in ref 6, polyamides were hydrogenated in DMSO, not THF. And in ref 6 also PA66 is hydrogenated to the diol and diamine, see entry 4 in table 2 in this ref. Not only 6-amino-1-hexanol. THF was used in the system presented in ref 20 of the manuscript. By the way, in ref 20, the products obtained are 1,6-hexamethylenediamine and 1,6-hexanediol in yields up to 78% for a model PA (see table 2 in ref 20). This should also be reflected here.

We thank the reviewer for pointing out this error, and Supplementary Table 1 now accurately represents the conditions reported in the reference, see Supplementary Table 1 (Reference 6) in response 2 above.

Reviewer #2 (Remarks to the Author):

The manuscript titled “Controlling the selectivity of the hydrogenolysis of polyamides catalysed by ceria supported metal nanoparticles” reports the selective hydrogenolysis of polyamide models and difficult-to-recycle ‘real-world’ polyamide waste using M/CeO₂ catalysts with high activities. Density functional theory has been used to rationalize selectively, and the effect of hydrogenolysis conditions on catalyst activity and selectivity have been investigated. Overall, this is a very thorough, logical, and well-conducted study in an important field of research. Therefore, I recommend publication in Nature Communications after some minor corrections and suggestions detailed below:

We thank the reviewer for their supportive comments and positive feedback.

1. The resolution of the TOC and images throughout the manuscript and supplementary information could be improved.

We have provided higher quality vector graphics files to enhance the clarity of all figures.

2. Introduction: please define PA-6 as Nylon-6 upon first use.

This definition has now been added the first time PA-6 is mentioned:

Herein, we report the study of the catalytic hydrogenolysis of *N*-hexylhexamide (which serves as a model substrate for PA-6, also known as Nylon-6, one of the most produced PAs²) using platinum-group metal (Ru, Rh, Pd, Ir, and Pt) nanoparticles (NPs) supported on ceria (CeO₂).

3. The catalyst characterization and theoretical studies relate very nicely to the experimental data, with detailed explanations provided. Have any attempts been made to look at the acidity of the catalysts? Differences in acidity could potentially explain the differences in activity observed for the Ru/MxO_y catalysts. Additionally, H₂ chemisorption could also be used to quantify the surface area and dispersion of the metal NPs on the surface. Note: these are suggestions, not requirements for publication.

We appreciate the suggestions, but since the mechanism is dominated by the metal nanoparticles we decided not to perform the additional experiments, however, we have included a possible reason for the differences in activity, also see response to point 4 below:

The hydrogenolysis activity of Ru supported on CeO₂ is the highest among all the Ru catalysts (Supplementary Fig. 11), possibly due to a higher tolerance to water and ammonia formed during the reaction^{49,57}.

4. The catalyst screening section states: “A control reaction was performed to determine the thermal stability of *N*-hexylhexanamide under the reaction conditions. The control reaction indicates that the substrate undergoes low levels (6%) of thermal decomposition, confirming that the observed hydrogenolysis activity was due to the activity of the metal catalysts.” The authors should consider mentioning here that 24.4% conversion is observed with just the CeO₂ support,

with the conversion increasing when metal NPs are on the surface. The current sentence implies that all the observed activity is from the metal NPs, whereas some activity may also come from the support.

We have rewritten the paragraph to also account for the activity of the support:

Control reactions were performed to determine the thermal stability of N-hexylhexanamide under the reaction conditions, which undergoes low levels (6%) of thermal decomposition in the absence of catalyst. In the presence of CeO₂, 24% conversion to non-alkane products is observed (Supplementary Table 4). These observations confirm that hydrogenolysis of PAs to alkanes with the M/CeO₂ catalysts is primarily due to the activity of the supported metal NPs⁵⁰.

5. Fig 2b. The scheme is missing H₂O as a product on the first arrow.

We have included the initial loss of H₂O as requested, see revised Figure 2b below:

6. Catalyst screening: Reference Supplementary Figure 2c when discussing PdCl₂.

We have added a reference to the figure as requested:

The lower activity of Pd/CeO₂ may be attributed to the higher proportion of PdCl₂ species, as revealed by XPS analysis (Supplementary Fig. 2c).

7. Figures S11 and S12: add the overall conversion to the plots (similar to Figures 2 and 3 in the manuscript).

Overall conversions have been added to these plots:

Supplementary Figure S11:

Supplementary Figure S12:

8. It's great that these catalysts are able to break down real-world plastic waste, PA blends, and PA composites. It would be beneficial to mention some of the uses and value of the products (methane, n-hexane, n-dodecane etc.) in the conclusions.

See response to reviewer 1, comment 5, where we have elaborated on the uses and value of the products.

Reviewer #3 (Remarks to the Author):

Revisions:

The manuscript demonstrates the hydrogenolysis of various PAs catalysed by platinum-group metal nanoparticles (NPs) supported on ceria (CeO₂). Ru/CeO₂ was found to be the most active in catalysing hydrogenolysis of PA materials rapidly into water, methane and ammonia, whereas Pt/CeO₂ yields hydrocarbons retaining the carbon backbone of the parent PA. The DFT computations were also performed to rationalize the observed selectivity. The work is meaningful, and can be accepted after a major revision. My detailed comments are as follows:

We thank the reviewer for their positive comments.

1. In page 4, the authors mentioned that “there have been limited studies exploring the cleavage of carbon–heteroatom bonds in PAs”. However, there are numerous experimental and theoretical studies that have been reported about the homogeneous hydrogenolysis of PA and polyamides. Some literatures (e.g. Chem. Sci., 2021, 12, 10590; JACS Au 2021, 1, 517–524; Inorg. Chem. 2022, 61, 14662–14672) should be introduced.

Two of the three papers mentioned are concerned with polyurethanes which contain weaker C-N bonds than polyamides. Since the focus of our manuscript is on polyamides and not polyurethanes we have only cited the work on the former, although we can of course extend the introduction to polyurethanes if the reviewer believes it is important. We have referenced (in the introduction) two pioneering works involving the homogeneous hydrogenation of polyamides (Reference 19 and 20).

In addition, we have put more emphasis on the lack of heterogeneous catalysts for PA hydrogenolysis in the introduction, see response to reviewer 1, comment 3.

2. What is more, the advantages of Ru/CeO₂ and Pt/CeO₂ over the homogeneous catalysts must be described in detail.

We have clarified the advantages of Ru/CeO₂ and Pt/CeO₂ over the homogeneous catalysts in the discussion:

Beyond the high selectivity the Ru/CeO₂ and Pt/CeO₂ catalysts, they are easy to prepare, highly robust, as evidenced by the stability of their conversion and product selectivity over multiple reaction cycles and allow facile separation and reuse. Moreover, solvent is not required as the reaction is performed above the melting point of the PA substrate with solvent being formed as the reaction progresses, which possibly contributes to the stability of the catalysts by preventing coking.

3. In Fig. 2, the N-hexylhexanamide undergoes the C-O hydrogenolysis to form a secondary amine intermediate, and the C-N hydrogenolysis to form a N-terminal amine. However, amides are widely reported to undergo the hydrogenolysis to provide N-terminal amines and alcohols (ACS Catal. 2020, 10, 5511–5515; J. Am. Chem. Soc. 2019, 141, 12962–12966; Nat. Commun. 2015, 6, 6859), and the secondary amine intermediates are not involved. The mechanism in Fig. 2 could not be right.

The original mechanism in Fig. 2 did not include certain intermediates, which led to the impression that it was incorrect. Hence, to clarify the mechanistic hypothesis, and the supporting experimental data, we have added intermediate steps to the reaction mechanism, see revised Fig. 2 above, and added a reference to support the proposed mechanism, *Chem. Rev.* 2014, 114, 5477–5510 as Reference 51:

Initial C-O cleavage via a hemiaminal intermediate is considered to be the predominant reaction pathway for most heterogeneous amide hydrogenation catalysts, with the absence of O-containing products in the product distribution supporting this mechanism⁵¹.

4. The discussion of chemoselectivity is based on the reactions of CH₃CCH in scenario II of Fig. 4, but the theoretical study does not provide the pathway for the formation of CH₃CCH intermediate. Otherwise, the experiment should prove the formation of CH₃CCH.

The CH₃CCH species is an intermediate that will be formed after consecutive C-H cleavages. The DFT energy pathway of such formation had been reported in *Phys. Chem. Chem. Phys.*, 2011,13, 3257-3267 and was added as Ref 64. To clarify that it is an intermediate species, we rewrote the sentence:

The C₃ species may also undergo several C-H cleavages, to generate the low-H content intermediate CH₃CCH, in which both the α -C and β -C atoms are adsorbed on the metal surface (Scenario II), as reported in other studies^{31,64}.

REVIEWERS' COMMENTS

Reviewer #1 (Remarks to the Author):

In this revised manuscript, the authors addressed all comments from the three reviewers well and added the corresponding parts to the manuscript. Therefore, it can now be considered for publication in Nature Communications.

Reviewer #3 (Remarks to the Author):

My concern have been addressed, and I advised the publishment of the manuscript.